# Early-generated interneurons regulate neuronal circuit formation during early postnatal development

Chang-Zheng Wang[1], Jian Ma[2], Ye-Qian Xu[1], Shao-Na Jiang[1], Tian-Qi Chen[1], Zu-Liang Yuan[1], Xiao-Yi Mao[1], Shu-Qing Zhang[1], Lin-Yun Liu[1], Yinghui Fu[1]*, Yong-Chun Yu[1]*

[1]Jing'an District Centre Hospital of Shanghai, State Key Laboratory of Medical Neurobiology and MOE Frontiers Center for Brain Science, Institutes of Brain Science, Fudan University, Shanghai, China; [2]School of Life Sciences, Tsinghua-Peking Joint Center for Life Sciences, IDG/McGovern Institute for Brain Research, Tsinghua University, Beijing, China

**Abstract** A small subset of interneurons that are generated earliest as pioneer neurons are the first cohort of neurons that enter the neocortex. However, it remains largely unclear whether these early-generated interneurons (EGIns) predominantly regulate neocortical circuit formation. Using inducible genetic fate mapping to selectively label EGIns and pseudo-random interneurons (pRIns), we found that EGIns exhibited more mature electrophysiological and morphological properties and higher synaptic connectivity than pRIns in the somatosensory cortex at early postnatal stages. In addition, when stimulating one cell, the proportion of EGIns that influence spontaneous network synchronization is significantly higher than that of pRIns. Importantly, toxin-mediated ablation of EGIns after birth significantly reduce spontaneous network synchronization and decrease inhibitory synaptic formation during the first postnatal week. These results suggest that EGIns can shape developing networks and may contribute to the refinement of neuronal connectivity before the establishment of the adult neuronal circuit.
DOI: https://doi.org/10.7554/eLife.44649.001

*For correspondence:
fuyh@fudan.edu.cn (YF);
ycyu@fudan.edu.cn (Y-CY)

**Competing interests:** The authors declare that no competing interests exist.

## Introduction

γ-aminobutyric acid (GABA)-ergic inhibitory interneurons comprise ~20% of the neuronal population in the neocortex. A key feature of these interneurons is the incredibly rich diversity in their morphology, biochemical marker expression, electrophysiological properties and synaptic connectivity patterns (*Ascoli et al., 2008*), which allows them to dynamically sculpt neuronal activity and network oscillations both during development and upon maturation (*Liguz-Lecznar et al., 2016*; *Whittington and Traub, 2003*), and endow neural circuits with remarkable computational power (*Kepecs and Fishell, 2014*). Considerable evidence suggests that GABAergic interneurons play crucial roles in several aspects of neural circuit development, including circuit formation and maturation, and synaptic plasticity (*Anastasiades et al., 2016*; *Dehorter et al., 2017*; *Le Magueresse and Monyer, 2013*). For example, disruption of the early-born SST interneurons located in cortical layer 5/6 during the first postnatal week impedes the synaptic maturation of thalamocortical inputs onto infra-granular PV interneurons (*Tuncdemir et al., 2016*). A recent study has also shown that developmental dysfunction of VIP interneurons by deletion of ErbB4 from these cells causes long-term defects in excitatory and inhibitory cortical neurons and impairs sensory processing and perception (*Batista-Brito et al., 2017*). Disruption of the developing GABAergic neocortical inhibitory network has been implicated in neurodevelopmental disorders, including schizophrenia, epilepsy, and autism

(*Cobos et al., 2005*; *Lewis et al., 2005*; *Pizzarelli and Cherubini, 2011*). However, whether this specific subpopulation of interneurons can precisely regulate neocortical circuit development remains largely unknown.

The generation of neocortical interneurons begins at embryonic day (E) 9.5, peaks at E12 to E15 and ends at E18.5 in mice (*Batista-Brito and Fishell, 2009*; *Miyoshi et al., 2010*; *Miyoshi et al., 2007*). Each temporal cohort exhibits specific physiological properties based on their birthdate and has distinct functional roles in the neocortex (*Butt et al., 2005*; *Donato et al., 2015*). Among these temporal cohorts, accumulating evidence suggests that the earliest generated cohort is a unique subpopulation of interneurons (*Allene et al., 2012*; *Picardo et al., 2011*; *Tuncdemir et al., 2016*; *Villette et al., 2016*). Based on the preferential attachment rule that early emerging individuals in a network have a strong 'first-mover advantage' (*Barabasi and Albert, 1999*), it has long been postulated that early-generated interneurons (EGIns) may develop a subpopulation of functional hub neurons and play a key role in regulating neural development, neuronal network dynamics and circuit formation. Indeed, genetic fate mapping studies have shown that a subpopulation of EGIns in the developing hippocampus and entorhinal cortex displays high functional connectivity and serves as functional hub cells by exerting a powerful effect on network synchronization at the end of the first postnatal week (*Bonifazi et al., 2009*; *Cossart, 2014*; *Mòdol et al., 2017*; *Picardo et al., 2011*). In addition, GABAergic hub neurons are characterized by an exceptionally widespread axonal arborization and preferentially express somatostatin (*Cossart, 2014*; *Mòdol et al., 2017*; *Picardo et al., 2011*). While these pioneering studies provided crucial insights into the potential role of EGIns in neural development and maturation of entorhinal-hippocampal circuits, several questions still remain. For instance, in the neocortex, what characteristic features do EGIns have? Does perturbation of a single neocortical EGIn influence the spontaneous network synchronization during early postnatal stages? As a sparse cell population, do cortical EGIns play a role in the functional maturation of the neocortex?

To address these issues, we used an inducible genetic fate-mapping approach to selectively label EGIns and pseudo-random interneurons (pRIns) in the developing neocortex. We observed that EGIns display more mature electrophysiological and morphological properties and higher local synaptic connectivity than pRIns at early postnatal stages. Moreover, a subpopulation of EGIns in neocortical layer 5, but very few pRIns, could single-handedly influence network dynamics. Importantly, ablation of EGIns resulted in defects in spontaneous network synchronization and inhibitory synapse formation in the early postnatal neocortex. Our results thereby identify a role for these sparse EGIns in cortical circuit development during early postnatal stages.

## Results

### EGIns are predominantly comprised of SST-positive interneurons and located in deep neocortical layers

Dlx1/2 is a transcription factor that is transiently expressed in almost all forebrain interneurons as they become postmitotic (*Eisenstat et al., 1999*). To selectively label interneurons that are generated at specific embryonic stages, we created the inducible *Dlx1/2-CreER$^{+/-}$; Rosa26-EYFP$^{\pm}$* mouse line by crossing *Dlx1/2-CreER$^{+/-}$* driver line (*Batista-Brito et al., 2008*) with *Rosa26-EYFP* Cre-dependent reporter line. This genetic fate-mapping strategy allows for permanent labeling of GABA neurons by maternal tamoxifen administration at specific time points in both embryonic and postnatal stages (*Batista-Brito et al., 2008*; *Picardo et al., 2011*; *Villette et al., 2016*). To label EGIns and pRIns, we administered tamoxifen to pregnant *Dlx1/2-CreER$^{+/-}$; Rosa26-EYFP$^{\pm}$* mice at E9.5 (onset of neurogenesis for cortical interneurons) and E13.5 (a peak period of neurogenesis for cortical interneurons), respectively (*Figure 1A*). It should be noted that, as the control group, pRIns are born from E9.5 to E13.5 and mostly derived from MGE (*Batista-Brito and Fishell, 2009*; *Ghanem et al., 2007*; *Miyoshi et al., 2010*; *Miyoshi et al., 2007*; *Potter et al., 2009*). To confirm the temporal resolution of our fate-mapping approach, we administered a tamoxifen solution (0.06 mg/g) by oral gavage to pregnant *Dlx1/2-CreER$^{+/-}$; Rosa26-EYFP$^{\pm}$* mice at E13.5, followed by a single 5-bromo-2'-deoxyuridine (BrdU) i.p. injection at E15.5, and detected few BrdU$^+$/EYFP$^+$ cells in the somatosensory cortex at P6 (*Figure 1—figure supplement 1A and B*). This result indicates that the tamoxifen efficacy lasts no longer than two days. Similarly, 1.6 ± 0.3% of EYFP$^+$ cells (n = 20 sections from four

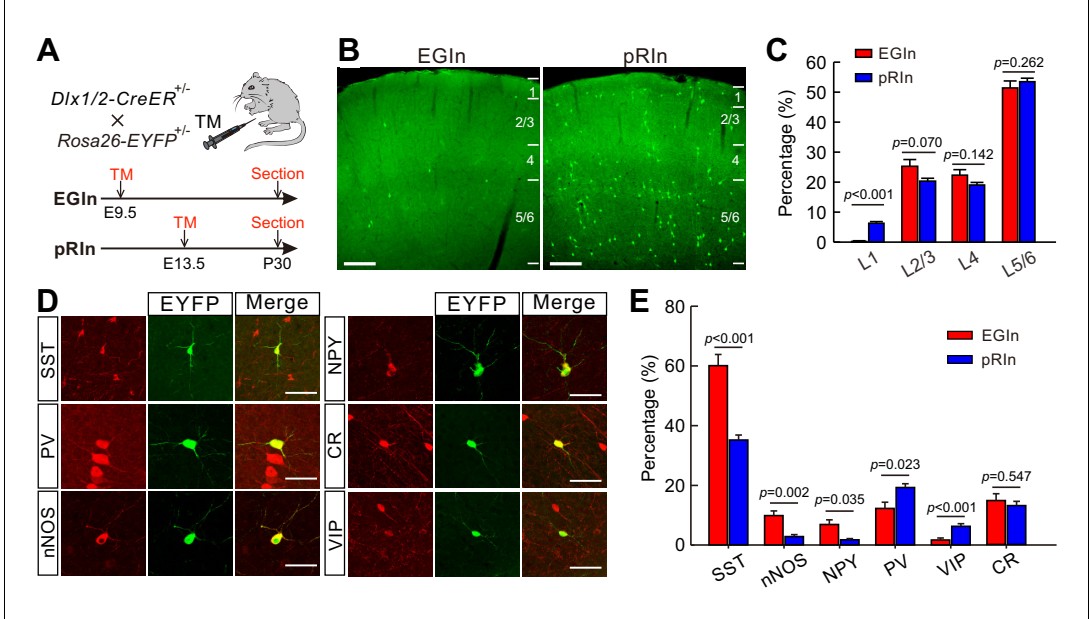

**Figure 1.** Laminar distribution and molecular marker expression of EGIns and pRIns in the somatosensory cortex at P30. (**A**) Schematic diagram representing the inducible transgenic strategy for labeling EGIns and pRIns. (**B**) Representative images showing the laminar distribution of EGIns (left) and pRIns (right). Scale bar, 200 μm. (**C**) Percentages of EGIns and pRIns located in different layers of the somatosensory cortex. (**D**) Sample images showing the co-expression of EYFP$^+$ neurons with SST, nNOS, NPY, PV, VIP and CR. Scale bar, 50 μm. (**E**) Percentages of EGIns and pRIns that were also positive for various molecular markers. Detailed statistical analysis, detailed data, and exact sample numbers are presented in *Figure 1—source data 1*. Error bars indicate mean ± SEM.

DOI: https://doi.org/10.7554/eLife.44649.002

The following source data and figure supplements are available for figure 1:

**Source data 1.** Detailed statistical analysis, detailed data, exact sample numbers, and *p* values in *Figure 1* and *Figure 1—figure supplements 2* and *3* and detailed cell densities of EGIns and pRIns.
DOI: https://doi.org/10.7554/eLife.44649.006

**Figure supplement 1.** Short-term fate mapping of Dlx1/2-CreER$^{+/-}$; Rosa26-EYFP$^{+/-}$ line.
DOI: https://doi.org/10.7554/eLife.44649.003

**Figure supplement 2.** Characterization of the original cells labeled by Dlx1/2-CreER at E9.5 and E13.5.
DOI: https://doi.org/10.7554/eLife.44649.004

**Figure supplement 3.** The percentages of sEGIns (layer 2/3) and dEGIns (layer 5/6) that express various molecular markers.
DOI: https://doi.org/10.7554/eLife.44649.005

mice) were positive for BrdU in the somatosensory cortex of *Dlx1/2-CreER$^{+/-}$; Rosa26-EYFP$^{\pm}$* mice at P5 that were gavaged with tamoxifen at E9.5 and injected BrdU at E13.5 (*Figure 1—figure supplement 1C and D*), suggesting that EGIns and pRIns are two temporally separated cohorts.

In addition, we characterized the precise cell types labeled by Dlx1/2-CreER at E9.5 and E13.5. 36 hr after tamoxifen administration, embryonic brain sections were stained with antibodies against OLIG2 and Ki67. In these experiments, OLIG2$^+$/Ki67$^+$, OLIG2$^-$/Ki67$^+$, and OLIG2$^-$/Ki67$^-$ cells corresponded to radial glial progenitors (RGPs), intermediate progenitors (IPs), and post-mitotic interneurons (INs), respectively (*Sultan et al., 2018*). We found the majority of Dlx1/2-EYFP$^+$ cells at E11 and E15 were OLIG2$^-$/Ki67$^-$ (100% at E11, n = 9 sections from three embryos; 99.5 ± 0.5% at E15, n = 12 sections from four embryos) (*Figure 1—figure supplement 2*), indicating that Dlx1/2-CreER line predominantly labels post-mitotic interneurons at the embryonic stage.

The populations of EGIns and pRIns were analyzed using immunohistochemical approaches at P30. The density of pRIns was approximately 14 times higher than that of EGIns in the neocortex (pRIns, 41.4 ± 2.0 cells/mm$^2$, n = 25 sections from six mice; EGIns, 3.0 ± 0.2 cells/mm$^2$, n = 28 sections from seven mice; *Figure 1—source data 1*), indicating that EGIns are a very sparse population of cortical interneurons. Of note, consistent with previous report (*Lim et al., 2018*; *Villette et al., 2016*), we found EGIns included approximately 10% of leaky EYFP$^+$ cells in immunohistochemical

cortical sections, but none in live cortical slices. We further characterized and compared the laminar distribution of EGIns and pRIns in the somatosensory cortex. Both EGIns and pRIns exhibited a location bias towards infragranular layers (layer 5–6, L5/6; *Figure 1B and C*). Moreover, the proportion of EYFP$^+$ cells in L2/3, L4 and L5/6 was comparable between EGIns and pRIns, although a greater proportion of pRIns was observed in L1 (*Figure 1B and C*). To examine the diversity of interneuron subtypes represented by EGIns and pRIns, we stained the EYFP$^+$ cells for a number of interneuron markers, including somatostatin (SST), parvalbumin (PV), neural nitric oxide synthase (nNOS), neuropeptide Y (NPY), calretinin (CR), and vasoactive intestinal peptide (VIP) (*Figure 1D*). In agreement with the temporal bias in the origins of the interneuron subgroups (*Hu et al., 2017*; *Miyoshi et al., 2010*; *Miyoshi et al., 2007*; *Rudy et al., 2011*),~60% of EGIns were positive for SST, and the proportion was much higher than that of pRIns (EGIns, 60.1 ± 3.7%, n = 7 mice; pRIns, 35.2 ± 1.7%, n = 6 mice; p<0.001, two-tailed unpaired *t*-test; *Figure 1E*). In addition, we found significantly more EGIns expressing nNOS (EGIns, 9.9 ± 1.6%, n = 7 mice; pRIns, 2.9 ± 0.6%, n = 6 mice; p=0.002, Mann Whitney *U* test; *Figure 1E*) and NPY (EGIns, 6.9 ± 3.7%, n = 7 mice; pRIns, 1.8 ± 0.4%, n = 6 mice; p=0.035, Mann Whitney *U* test; *Figure 1E*). In contrast, the proportions of pRIns expressing PV and VIP were significantly higher than those of EGIns (PV, 12.3 ± 2.1% for EGIns, n = 7 mice; 19.3 ± 1.3% for pRIns, n = 6 mice; p=0.023, Mann Whitney *U* test; VIP, 1.7 ± 0.7% for EGIns, n = 7 mice; 6.3 ± 0.9% for pRIns, n = 6 mice; p<0.001, Mann Whitney *U* test; *Figure 1E*). The proportion of CR-expressing cells was similar in EGIns and pRIns (EGIns, 14.8 ± 2.3%, n = 7 mice; pRIns, 13.1 ± 1.4%, n = 6 mice; p=0.547, two-tailed unpaired *t*-test; *Figure 1E*). Together, these results suggest that SST$^+$ interneurons are the main subpopulation among the earliest born cohort of interneurons and invade the deep layers of the cerebral cortex. Given that EGIns were observed in both superficial (L2/3, sEGIns) and deep (L5/6, dEGIns) layers, we further compared the biochemical marker expression between sEGIns and dEGIns. While the proportion of sEGIns expressing CR was significantly higher than that of dEGIns, the expressions of other biochemical markers were similar in sEGIns and dEGIns (*Figure 1—figure supplement 3*).

## Electrophysiological and morphological properties of EGIns and pRIns at the early postnatal stages

We next asked whether the electrophysiological and morphological characteristics of EGIns differed from those of pRIns at the early postnatal stages. Focusing on layer 5 of the somatosensory cortex, we performed whole-cell patch-clamp recordings of EYFP$^+$ EGIns and pRIns in acute in vitro cortical slices at P5–7 (*Figure 2A*). Recorded cells were filled with neurobiotin for *post hoc* morphological analysis (*Figure 2F*). Four electrophysiological features thoes described the intrinsic electrophysiological properties of neurons were analyzed, including action potential (AP) threshold, AP amplitude, AP width and input resistance. Although AP threshold and AP amplitude of EGIns were similar to those of pRIns, AP width and input resistance of EGIns were significantly lower than those of pRIns (*Figure 2B–E*). These results suggest that EGIns exhibit more mature electrophysiological properties compared with pRIns at the early postnatal stages. Moreover, we observed that dEGIns showed more mature electrophysiological properties than sEGIns at P5–7 (*Figure 2—figure supplement 1*).

Furthermore, we systematically analyzed the dendritic and axonal morphology of EGIns (n = 28) and pRIns (n = 30) at P5–7 (*Figure 2F*). Consistent with their electrophysiological properties, EGIns exhibited more mature morphological features than pRIns, such as longer total branch length, larger surface area and more node numbers in both dendrites and axons (*Figure 2G–L*). Similarly, dEGIns showed more mature morphological properties than sEGIns (*Figure 2—figure supplement 2*). Interestingly, there were no significant differences in electrophysiological and morphological properties between EGIns and pRIns at P15–20 (*Figure 2—figure supplements 3* and *4*). These results suggest that the electrophysiological and morphological features of EGIns compared to pRIns are maintained only for a short time in the developing neocortex.

## EGIns display more miniature postsynaptic currents than pRIns at early postnatal stages

Our electrophysiological and morphological results suggest that EGIns display more mature properties than pRIns during the first postnatal week. To test whether the electrophysiological and morphological properties correspond to the functional synaptic connectivity of EGIns, we performed whole-

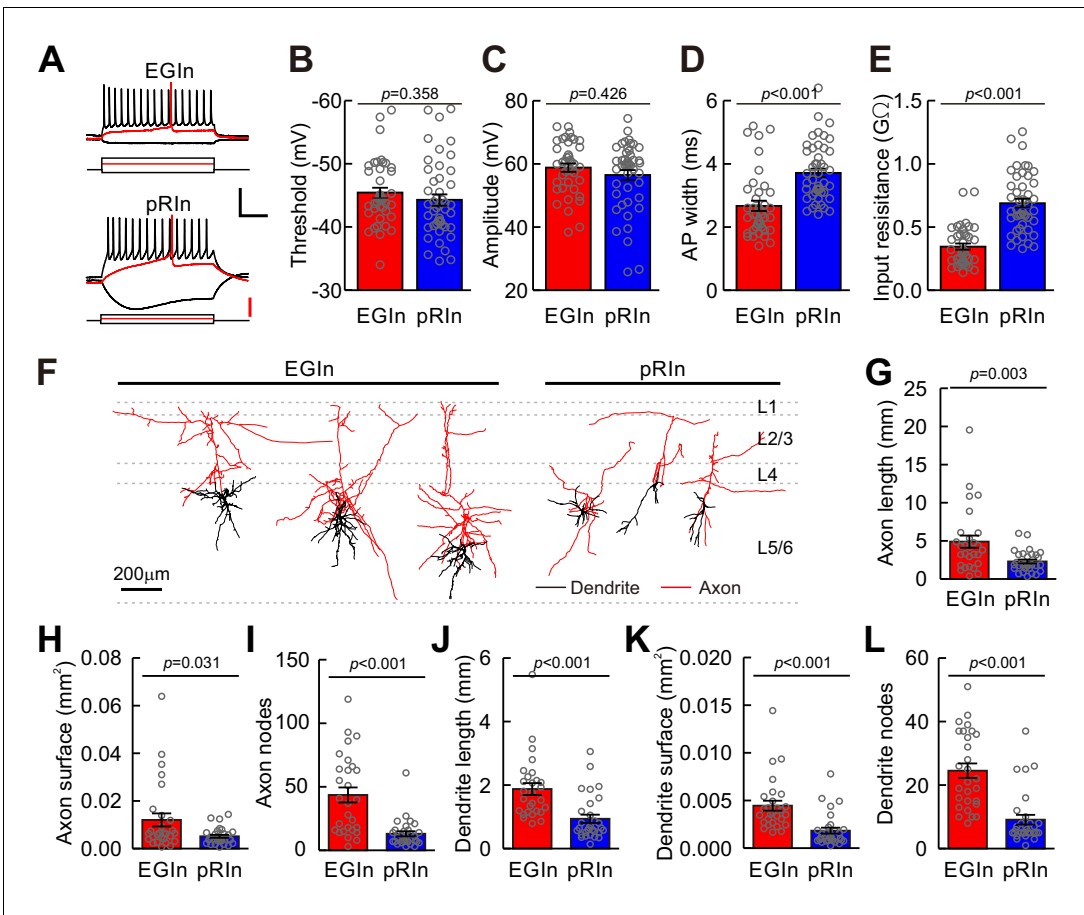

**Figure 2.** Electrophysiological and morphological properties of EGIns and pRIns at P5–7. (**A**) Representative traces showing voltage responses of EGIns (top) and pRIns (bottom) to step current injections. Red traces indicate the first evoked action potential. Scale bars: 40 mV (vertical, black), 200 pA (vertical, red), and 200 ms (horizontal, black). (**B–E**) Comparison of AP threshold (**B**), AP amplitude (**C**), AP width (**D**) and input resistance (**E**) between EGIns and pRIns. (**F**) Neurolucida reconstructions of EGIns (left 3 cells) and pRIns (right 3 cells) in layer 5/6 of the somatosensory cortex. Scale bar, 200 μm. (**G–L**) Comparison of axon length (**G**), axon surface (**H**), axon nodes (**I**), dendrite length (**J**), dendrite surface (**K**) and dendrite nodes (**L**) between EGIns and pRIns. Detailed statistical analysis, detailed data, and number of experiments are presented in the *Figure 2—source data 1*.
DOI: https://doi.org/10.7554/eLife.44649.007

The following source data and figure supplements are available for figure 2:

**Source data 1.** Detailed statistical analysis, detailed data, exact sample numbers, and *p* values in *Figure 2* and *Figure 2—figure supplement 1–4*.
DOI: https://doi.org/10.7554/eLife.44649.012
**Figure supplement 1.** Electrophysiological properties of sEGIns (layer 2/3) and dEGIns (layer 5/6) at P5–7.
DOI: https://doi.org/10.7554/eLife.44649.008
**Figure supplement 2.** Morphological properties of dEGIns and sEGIns at P5–7.
DOI: https://doi.org/10.7554/eLife.44649.009
**Figure supplement 3.** Electrophysiological properties of EGIns and pRIns at P15–20.
DOI: https://doi.org/10.7554/eLife.44649.010
**Figure supplement 4.** Morphological properties of EGIns and pRIns at P15–20.
DOI: https://doi.org/10.7554/eLife.44649.011

cell patch-clamp recordings from layer 5 EYFP⁺ EGIns and pRIns in the somatosensory cortex at P5–7 (*Figure 3A,B and C*). We examined both miniature excitatory postsynaptic currents (mEPSCs) and miniature inhibitory postsynaptic currents (mIPSCs) in the same neurons in the presence of tetrodotoxin (TTX, 5 μM) (*Allene et al., 2012*; *Oh et al., 2016*; *Yao et al., 2016*) (*Figure 3D*). We found

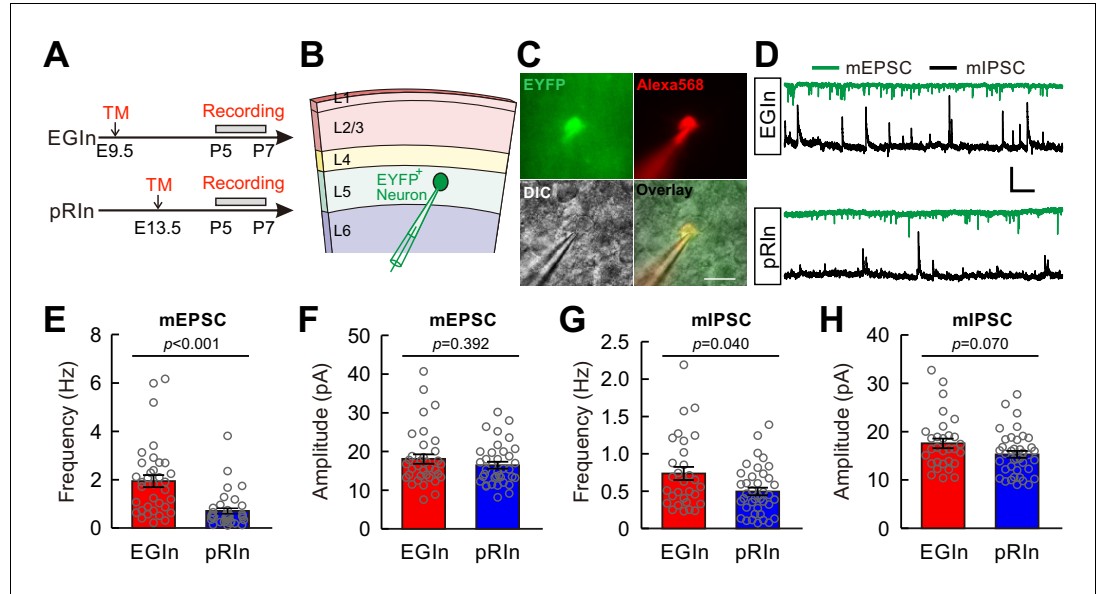

**Figure 3.** EGIns display more miniature postsynaptic currents than pRIns at early postnatal stages. (A) Schematic time schedule of electrophysiological recordings. (B) Schematic diagram of whole-cell recording of an EYFP+ neuron in layer 5. (C) Representative fluorescent (EYFP, EGIn; Alexa 568, recorded neurons), IR-DIC and merged images of whole- cell recording from an EYFP+ neuron in layer 5. Scale bar, 20 μm. (D) Representative traces of inward mEPSCs (green traces) and outward mIPSCs (black traces) recorded from EGIn and pRIn. Scale bars: 20 pA (vertical), 2 s (horizontal). (E and F) Histograms of mEPSC frequencies (E) and peak amplitudes (F) for EGIns and pRIns. (G and H) Histograms of mIPSC frequencies (G) and peak amplitudes (H) for EGIns and pRIns. Detailed statistical analysis, detailed data and number of experiments are presented in the *Figure 3—source data 1*.
DOI: https://doi.org/10.7554/eLife.44649.013

The following source data and figure supplement are available for figure 3:

**Source data 1.** Detailed statistical analysis, detailed data, exact sample numbers, and *p* values in *Figure 3* and *Figure 3—figure supplement 1*.
DOI: https://doi.org/10.7554/eLife.44649.015
**Figure supplement 1.** Comparison of mEPSCs and mIPSCs between dEGIns and sEGIns at P5–7.
DOI: https://doi.org/10.7554/eLife.44649.014

that the frequency of mEPSCs and mIPSCs of EGIns was significantly higher than that of pRIns (EGIns, $1.9 \pm 0.3$ Hz for mEPSCs, n = 35 cells, $0.7 \pm 0.1$ Hz for mIPSCs, n = 30 cells; pRIns, $0.7 \pm 0.1$ Hz for mEPSCs, n = 37 cells, $0.5 \pm 0.1$ Hz for mIPSCs, n = 39 cells; p<0.001 for mEPSCs, p=0.040 for mIPSCs, Mann-Whitney *U* test; *Figure 3E and G*), while the peak amplitudes of mEPSCs and mIPSCs were indistinguishable between EGIns and pRIns (EGIns, $18.1 \pm 1.2$ pA for mEPSCs, n = 35 cells, $17.6 \pm 1.0$ pA for mIPSCs, n = 30 cells; pRIns, $16.5 \pm 0.9$ pA for mEPSCs, n = 37 cells, $15.3 \pm 0.7$ pA for mIPSCs, n = 39 cells; p=0.392 for mEPSCs, p=0.070 for mIPSCs, Mann-Whitney *U* test; *Figure 3F and H* ). These results suggest that EGIns receive more excitatory and inhibitory synaptic inputs than pRIns at early postnatal stages. In addition, we observed the frequency of mEPSCs and mIPSCs and the peak amplitude of mIPSCs were comparable between dEGIns and sEGIns, while the peak amplitude of mEPSCs of dEGIns was significantly higher than that of sEGIns (*Figure 3—figure supplement 1*).

## EGIns form higher synaptic connectivity than pRIns at early postnatal stages

To directly assess synaptic connectivity, we performed dual whole-cell patch-clamp recordings to simultaneously record from an EYFP+ EGIn or an EYFP+ pRIn and an adjacent pyramidal cell (PC) whose cell body was within ~100 μm in layer 5 of the somatosensory cortex at P5–7 (*Figure 4A,B and C*). Synaptic connections were probed by evoking unitary postsynaptic currents with a single

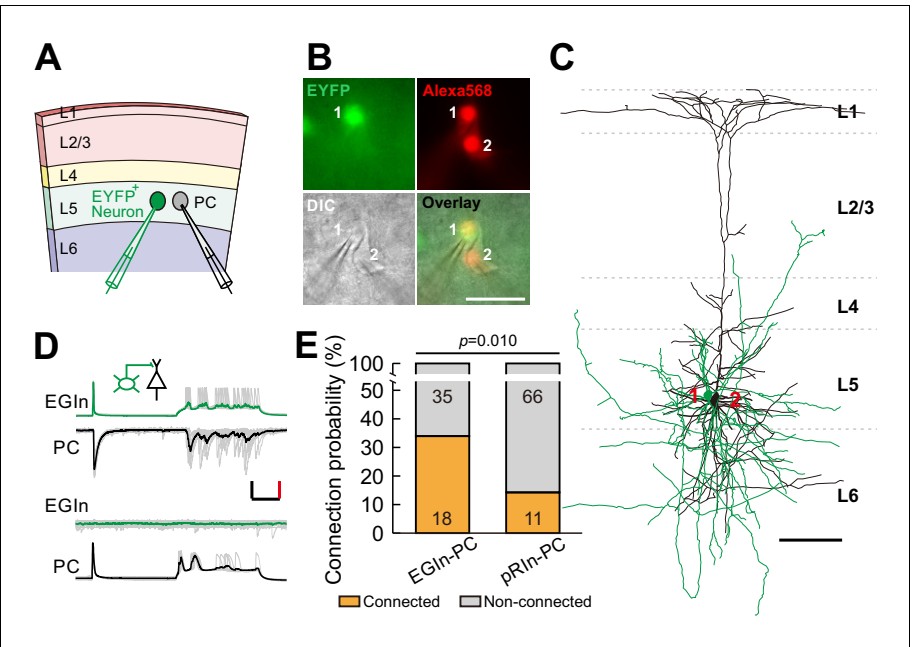

**Figure 4.** EGINs exhibit higher synaptic connectivity than pRIns at P5–7. (**A**) Schematic diagram represents dual patch-clamp recording of an EYFP$^+$ neuron and a neighboring PC in layer 5. (**B**) Representative fluorescent (EYFP, EGIn; Alexa 568, recorded neurons), IR-DIC and merged images of dual patch-clamp recording from an EYFP$^+$ neuron and a neighboring PC. Cell one is an EYFP$^+$ EGIn and cell two is a neighboring PC. Scale bar, 50 μm. (**C**) Reconstructed morphology of the two neurons patched in (**B**). Reconstructed EGIn is shown in green and PC is shown in black. Scale bar, 100 μm. (**D**) Representative traces showing an EGIn exert unidirectional chemical synapse onto a neighboring PC. The green (EGIn) and black (PC) lines indicate the average traces. Inset indicates unidirectional synaptic input from an EGIn to a PC. Scale bars: 50 pA (vertical, black), 50 mV (vertical, red), and 100 ms (horizontal, black). (**E**) Proportion of synaptic connections between EGIns and pyramidal cells and between pRIns and pyramidal cells.

DOI: https://doi.org/10.7554/eLife.44649.016

action potential (at least ten trials) triggered in the presynaptic neurons (*Figure 4D*). We found ~34.0% (18 out of 53) of EGIns formed unidirectional or bidirectional synaptic connections with excitatory neurons. The proportion of synaptic connections in EGIn-PC pairs was significantly higher than in pRIn-PC pairs (pRIn-PC pairs, 14.3%, 11 out of 77; two-tailed Fisher's exact test, p=0.010; *Figure 4E*). Together, these results suggest that EGIns have greater synaptic connectivity than pRIns at the early postnatal stages.

## A subpopulation of EGIns can single-handedly influence network dynamics

Spontaneous synchronous network activity has been found throughout the developing neocortex, and plays a critical role in neocortical development (*Bando et al., 2016*; *Kasyanov et al., 2004*; *Kirkby et al., 2013*; *Mohajerani and Cherubini, 2006*; *Vargas et al., 2013*; *Voigt et al., 2005*). Thus, we speculated that EGIns may modulate neocortical synchronous activity, which in turn shapes neuronal development and synapse formation. To test this hypothesis, we first examined whether a single EGIn could, when stimulated, influence synchronous network activity in the neocortex. In the immature neocortex, giant depolarizing potentials (GDPs) represent a primordial form of synchrony between neurons (*Allène et al., 2008*; *Rheims et al., 2008*). We performed dual whole-cell patch-clamp recordings to simultaneously record GDPs from an EYFP$^+$ EGIn or an EYFP$^+$ pRIn in current-clamp mode and an adjacent excitatory neuron in voltage-clamp mode in layer 5 of the somatosensory cortex at P5–7 (*Figure 5A*). As previously reported (*Ito, 2016*; *Picardo et al., 2011*; *Wester and McBain, 2016*), GDPs were identified by simultaneously occurring large membrane depolarizations in EYFP$^+$ cells and inward currents in excitatory neurons, lasting several hundreds of

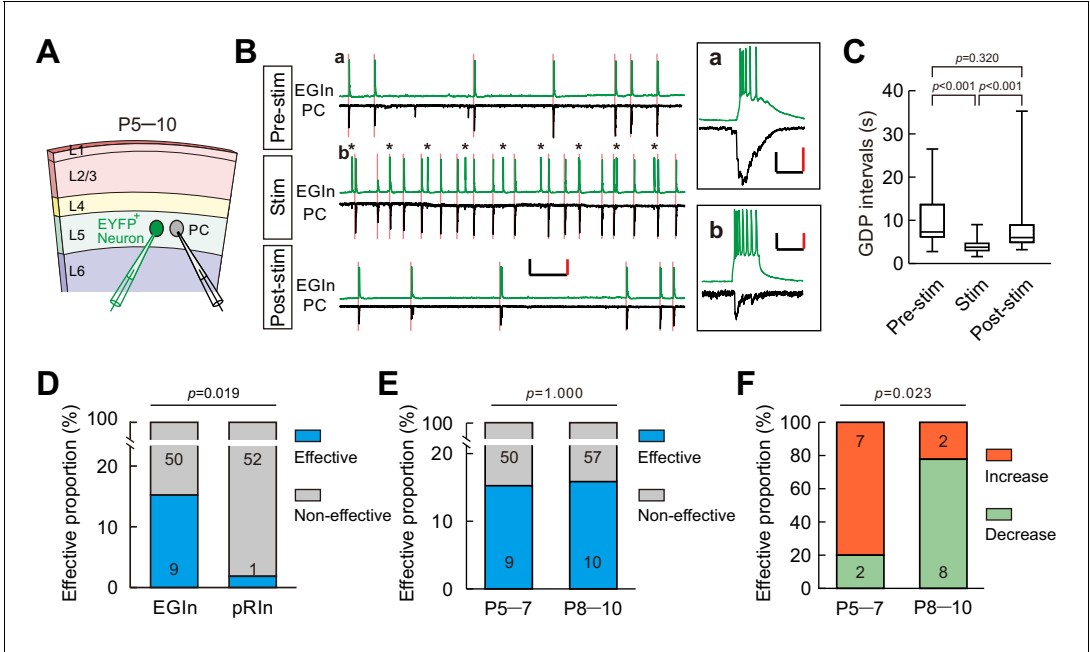

**Figure 5.** A subpopulation of EGIns can single-handedly influence network dynamics. (**A**) Schematic diagram showing a dual patch-clamp recording to test whether stimulating an EYFP⁺ neuron can influence network dynamics. (**B**) Representative traces indicate that stimulating an EYFP⁺ EGIn significantly increased the frequency of GDPs. Spontaneous activities were recorded from an EGIn and a pyramidal cell (PC) during different stimulus conditions. Scale bars: 200 pA (vertical, black), 40 mV (vertical, red), and 10 s (horizontal, black). Inset (a) showing a representative enlarged GDP recorded in EGIn and PC during pre-stimulus condition. Scale bars: 100 pA (vertical, black), 30 mV (vertical, red), and 200 ms (horizontal, black). Inset (b) showing synaptic transmission from EGIn to PC. Scale bars: 40 pA (vertical, black), 20 mV (vertical, red), and 200 ms (horizontal, black). Pink lines indicate the onsets of GDPs and asterisks indicate the 0.1 Hz, 200 ms current stimulation to EGIn. (**C**) Quantification of GDP intervals recorded in (**B**). (**D**) Proportion of EGIns that alter GDP frequency was significantly higher than that of pRIns at P5–7. Two-tailed Fisher's exact test, p=0.019. (**E**) Proportion of EGIns that alter GDP frequency exhibited no significant difference between P5–7 and P8–10. Two-tailed Fisher's exact test, p=1.000. (**F**) Proportion of EGIns that increased or decreased GDP frequency were compared between P5–7 and P8–10. Two-tailed Fisher's exact test, p=0.023. Detailed statistical analysis, detailed data, and exact sample numbers are presented in the *Figure 5—source data 1*. Error bars indicate mean ± SEM.

DOI: https://doi.org/10.7554/eLife.44649.017

The following source data and figure supplement are available for figure 5:

**Source data 1.** Detailed statistical analysis, detailed data, exact sample numbers, and *p* values in *Figure 5* and *Figure 5—figure supplement 1*.
DOI: https://doi.org/10.7554/eLife.44649.019

**Figure supplement 1.** Samples of decreasing and unaltered GDP frequency upon EGIn stimulation.
DOI: https://doi.org/10.7554/eLife.44649.018

milliseconds (*Figure 5B* inset *a*). To test the ability of single cells to influence GDPs, we stimulated a recording EYFP⁺ EGIn (200 ms pulse) to generate the burst firing of action potentials every ten seconds for 3 min (*Figure 5B*). A cell was considered to significantly affect network dynamics by statistically different distributions of GDP intervals in pre-stimulus conditions, during stimulation, and in post-stimulus conditions (Kruskal-Wallis test with Dunn's Multiple Comparison test, p<0.001; pre-stim vs stim, p<0.001; stim vs post-stim, p<0.001; pre-stim vs post-stim, p=0.320; *Figure 5C*). The typical samples of increasing GDP frequency, decreasing GDP frequency and non-alteration of GDP frequency upon EGIn stimulation are presented in *Figure 5B and C*, *Figure 5—figure supplement 1A and B* (one-way ANOVA with *post-hoc* Tukey HSD test, $F_{(2,34)}$=4.951, p=0.013; pre-stim vs stim, p=0.017; stim vs post-stim, p=0.023; pre-stim vs post-stim, p=0.900), *Figure 5—figure supplement 1C and D* (Kruskal-Wallis test, p=0.486), respectively. In some cases, APs in EYFP⁺ cells could induce postsynaptic inward currents in excitatory neurons (*Figure 5B* inset *b*). We found ~15.3% of EGIns could significantly affect GDP frequency (*Figure 5D*). In contrast, a lone pRIn exhibited significant effect on GDP frequency (EGIn, 9 out of 59; pRIn, 1 out of 53; two-tailed Fisher's exact test, p=0.019; *Figure 5D*).

In the developmental time window during which GDPs occur, GABAergic neurotransmission undergoes a functional switch from excitatory to inhibitory (*Ben-Ari, 2014*; *Ben-Ari et al., 2007*; *Dehorter et al., 2012*). Therefore, we further compared the impacts of EGIns on GDPs between two sequential postnatal periods: P5-P7 and P8-P10, and found that the proportion of EGIns that significantly affect GDP frequency was comparable between P5–5 and P8–10 (P5–7, 9 out of 59; P8–10, 10 out of 67; two-tailed Fisher's exact test, p=1.000; *Figure 5E*). Interestingly, we observed that EGIns at P5–7 preferentially increased GDP frequency when stimulated, whereas EGIns at P8–10 preferentially decreased GDP frequency when stimulated (P5–7, 7 cells showed increase versus 2 cells showed decrease; P8–10, 2 cells showed increase versus 8 cells showed decrease; two-tailed Fisher's exact test, p=0.023; *Figure 5F*). These observations indicate that the impact of EGIns on GDPs undergoes a switch from excitatory to inhibitory, which might correlate with the excitation-to-inhibition switch of GABAergic action.

## Ablation of sparse EGIns impairs spontaneous network synchronization and inhibitory synaptic formation at early postnatal stages

We next sought to address the importance of EGIns in cortical development by disrupting the EGIn population. To conditionally ablate EGIns or pRIns, we generated *Dlx1/2-CreER$^{+/-}$; Rosa26-iDTR$^\pm$* line by crossing *Dlx1/2-CreER$^{+/-}$* driver line with *Rosa26-iDTR* line (*Arruda-Carvalho et al., 2011*; *Buch et al., 2005*). This led to selective expression of diphtheria toxin receptor (DTR) in EGIns or pRIns after tamoxifen administration at E9.5 or E13.5, which allowed us to persistently tag infected neurons for subsequent ablation (*Figure 6A*). To ablate equal numbers of EGIns and pRIns, a relatively low-dose tamoxifen was administered at E13.5. Pups were then intraperitoneally injected with diphtheria toxin (DT) three times a day (once every 8 hr) from P2–4 (*Figure 6A*). 24 hr after the last DT administration (P5), we found the densities of both EGIns and pRIns in DT-treated mice were reduced to ~5% of that seen in saline-treated mice in the somatosensory neocortex (*Figure 6B and C*). These data demonstrate that DT treatment effectively ablates EGIns and pRIns in the neocortex at early postnatal stages. To further explore whether DT administration leads to non-Cre-expressing cell death, we labeled dying cells with caspase-3 and compared their densities between DT-injected CD1 mice and saline-injected CD1 mice at P5. We found the density of caspase-3-positive cells in the somatosensory neocortex was similar between DT-injected CD1 mice and saline-injected CD1 mice (*Figure 6—figure supplement 1*), indicating that the effect of DT is specific for Cre-expressing cells. In addition, compared with the density of GABAergic interneurons in the somatosensory neocortex of GAD67-GFP (Δneo) transgenic mice, we estimated the proportion of ablated neurons among total interneurons to be less than 0.4%.

We then asked whether EGIn ablation would have an effect on spontaneous network synchronization and synaptic connectivity at early postnatal stages. GDP responses were recorded from excitatory neurons in voltage-clamp mode (−70 mV holding potential) in layer 5 of the somatosensory cortex at P5–7 (*Figure 6D*). The average GDP frequencies were compared between EGIn DT-injected mice (EGIn-DT), pRIn DT-injected mice (pRIn-DT) and wild-type DT-injected mice (*Dlx1/2-CreER$^{-/-}$; Rosa26-iDTR$^{+/-}$*, WT-DT, tamoxifen administration at E9.5). We observed that the average GDP frequency in EGIn-DT mice was significantly lower than in pRIn-DT and WT-DT mice (EGIn-DT, 0.071 ± 0.009 Hz, n = 63 cells from seven mice; pRIn-DT, 0.092 ± 0.008 Hz, n = 43 cells from four mice; WT-DT, 0.102 ± 0.008 Hz, n = 58 cells from seven mice; EGIn-DT versus pRIn-DT, p=0.016, EGIn-DT versus WT-DT, p=0.034, Kruskal-Wallis test with Dunn's Multiple Comparison test; *Figure 6E*). In contrast, the average GDP frequency exhibited no significant difference between pRIn-DT and WT-DT mice (p=0.493, Kruskal-Wallis test; *Figure 6E*). These results indicate that EGIns contribute to spontaneous neocortical network synchronization at early postnatal stages. Moreover, in contrast with pRIn-DT and WT-DT mice, pyramidal neurons in layer 5 of the somatosensory cortex of EGIn-DT mice exhibited normal morphological and intrinsic electrophysiological properties at P5–7 (*Figure 6—figure supplements 2* and *3*). We further recorded mEPSCs and mIPSCs in layer five pyramidal neurons of the somatosensory cortex at P5–7 and compared them across EGIn-DT, pRIn-DT and WT-DT mice (*Figure 6F*). We found that the frequency of mIPSCs in EGIn-DT mice was significantly lower than in pRIn-DT and WT-DT mice (*Figure 6I*), whereas the frequency of mEPSCs and the peak amplitudes of mEPSCs and mIPSCs were indistinguishable between EGIn-DT, pRIn-DT and WT-DT mice (*Figure 6G,H and J*). These data suggest that EGIns are critical for proper spontaneous network synchronization and inhibitory synaptic transmission in the early postnatal neocortex.

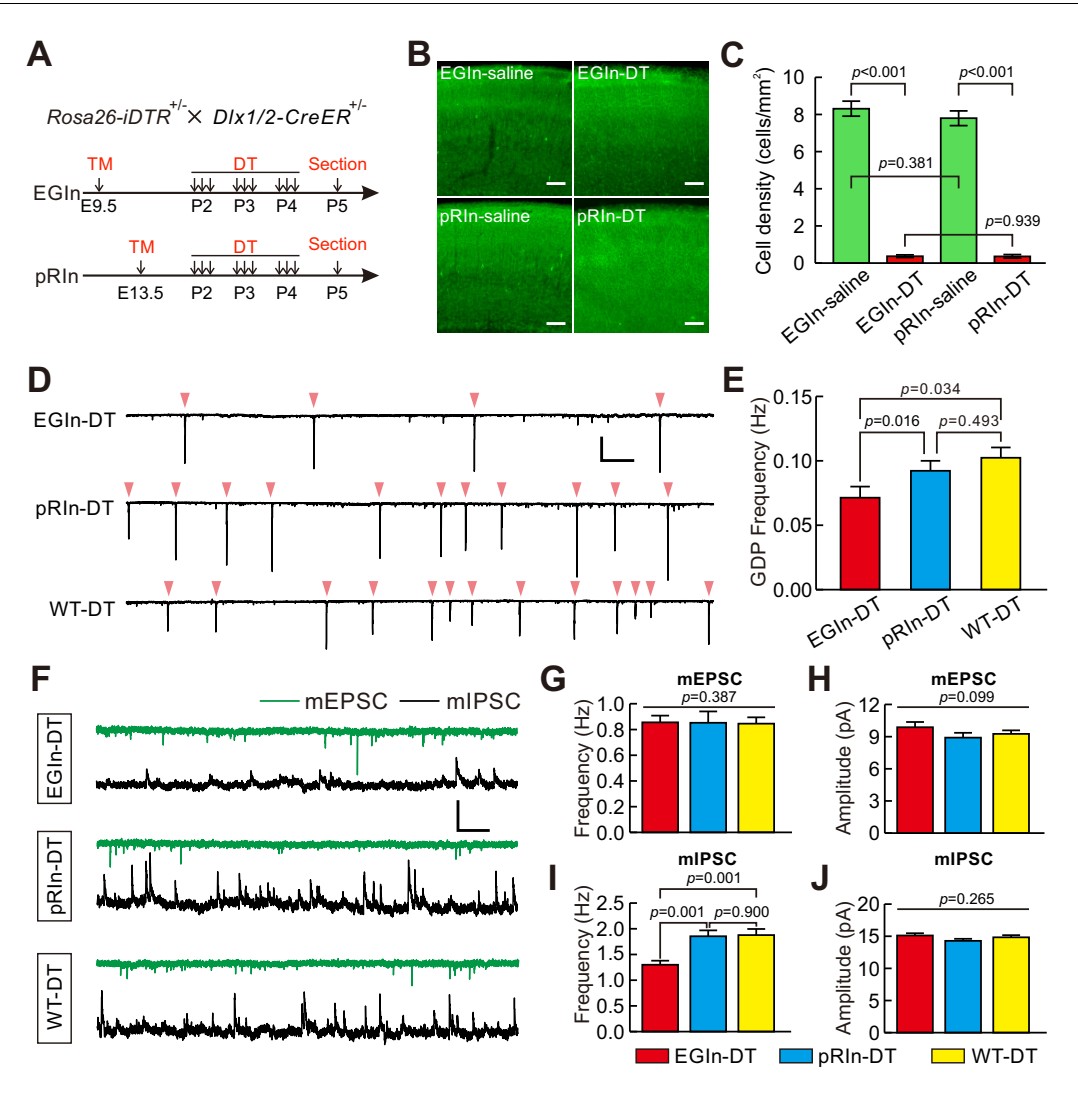

**Figure 6.** Ablation of EGIns reduced neocortical spontaneous network synchronization and the frequency of mIPSCs at early postnatal stages. (A) Schematic diagram representing ablation of EGIns and pRIns after P2. (B) Sample images showing DT administration can significantly reduce cell density of EGIns and pRIns at P5. Scale bar, 100 µm. (C) Quantitative analysis of DT ablation efficiency. (D) Sample traces showing GDPs recorded from EGIn DT-injected mice (top), pRIn DT-injected mice (middle) and wild-type DT-injected mice (bottom). Pink arrowheads indicate onsets of synchronized activities. Scale bars: 200 pA (vertical), 10 s (horizontal). (E) Quantitation of GDP frequencies in EGIn DT-injected mice, pRIn DT-injected mice and wild-type DT-injected mice. (F) Representative traces of inward mEPSCs (green traces) and outward mIPSCs (black traces) recorded in layer five pyramidal neurons in EGIn DT-injected mice, pRIn DT-injected mice and wild-type DT-injected mice. Scale bars: 20 pA (vertical), 1 s (horizontal). (G–J) Histograms of the frequencies (G) and amplitudes (H) of mEPSCs, and the frequencies (I) and amplitudes (J) of mIPSCs in EGIn DT-injected mice, pRIn DT-injected mice and wild-type DT-injected mice. Detailed statistical analysis, detailed data, and exact sample numbers are presented in the *Figure 6—source data 1*. Error bars indicate mean ± SEM.

DOI: https://doi.org/10.7554/eLife.44649.020

The following source data and figure supplements are available for figure 6:

**Source data 1.** Detailed statistical analysis, detailed data, exact sample numbers, and *p* values in *Figure 6* and *Figure 6—figure supplement 1–3*.

DOI: https://doi.org/10.7554/eLife.44649.024

**Figure supplement 1.** The effect of DT is specific for Cre-expressing cells.

DOI: https://doi.org/10.7554/eLife.44649.021

*Figure 6 continued on next page*

*Figure 6 continued*

**Figure supplement 2.** Ablation of EGIns did not change the intrinsic electrophysiological properties of layer five pyramidal cells at P5–7.

DOI: https://doi.org/10.7554/eLife.44649.022

**Figure supplement 3.** Ablation of EGIns did not change morphological properties of layer five pyramidal cells at P5–7.

DOI: https://doi.org/10.7554/eLife.44649.023

---

Although we found EGIn ablation reduced the frequency of mIPSCs in layer five pyramidal neurons, the impact might arise from the fact that EGIns are neurons with higher connectivity than pRIns at early postnatal stages, thus ablating them leads to a dramatic reduction of inhibition onto pyramidal cells. To exclude this possibility, we performed dual whole-cell patch-clamp recordings to simultaneously record a layer five interneuron (EYFP-negative, non-EGIns) and a nearby pyramidal cell at P5–7, and compared the connection probability and strength of unitary inhibitory postsynaptic currents (uIPSCs) from interneuron to pyramidal cell between EGIn-ablated mice (*Dlx1/2-CreER$^{+/-}$; Rosa26-iDTR$^{\pm}$* line; DT injected) and EGIn-EYFP mice (*Dlx1/2-CreER$^{+/-}$; Rosa26-EYFP$^{\pm}$* line; DT injected) (*Figure 7A,B and C*). Pyramidal cells and interneurons were identified with fluorescent

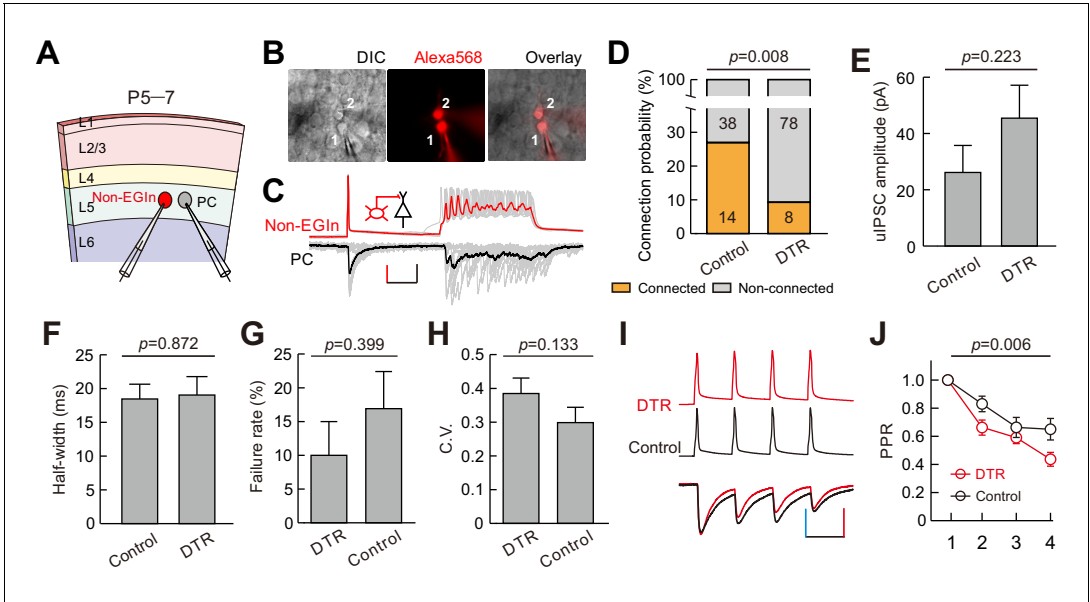

**Figure 7.** EGIns ablation altered synaptic formation and presynaptic transmitter release from non-EGIns to PCs at early postnatal stage. (**A**) Schematic diagram represents dual patch-clamp recording of an EYFP⁻ non-EGIn and a neighboring PC in layer 5. (**B**) Representative fluorescent (Alexa 568, recorded neurons), IR-DIC and merged images of dual patch-clamp recording from a non-EGIn and a neighboring PC. Cell one is a non-EGIn and cell two is a neighboring PC. (**C**) Representative traces showing a non-EGIn exert unidirectional chemical synapse onto a neighboring PC. The red (non-EGIn) and black (PC) lines indicate the average traces. Inset indicates unidirectional synaptic input from a non-EGIn to a PC. Scale bars: 30 pA (vertical, black), 30 mV (vertical, red), and 100 ms (horizontal, black). (**D**) Proportion of non-EGIns→PCs synaptic connections between EGIn-EYFP mice (control) and EGIn-ablated mice (DTR). (**E–F**) Quantification of the peak amplitude (**E**) and half-width (**F**) of non-EGIns→PCs uIPSCs between EGIn-EYFP mice (control) and EGIn-ablated mice (DTR). (**G–H**) Quantification of failure rate (**G**) and the coefficient of variation (C.V.) (**H**) of non-EGIns→PCs synaptic transmission. (**I**) Amplitude-scaled overlay of paired-pulse ratio (PPR) responses in non-EGIns→PCs connections between EGIn-EYFP mice (control) and EGIn-ablated mice (DTR). Red, DTR; black, control. Scale bars: 60 pA (vertical blue), 60 mV (vertical, red), and 50 ms (horizontal). Four presynaptic action potentials were evoked at 20 Hz. (**J**) The normalized peak amplitude of non-EGIns→PCs uIPSCs showed short-term depression, and significant difference in PPR was found between EGIn-EYFP mice (control) and EGIn-ablated mice (DTR). Detailed statistical analysis, detailed data, and exact sample numbers are presented in the *Figure 7—source data 1*. Error bars indicate mean ± SEM. Figure Supplement and Source data.

DOI: https://doi.org/10.7554/eLife.44649.025

The following source data is available for figure 7:

**Source data 1.** Detailed statistical analysis, detailed data, exact sample numbers, and *p* values in *Figure 7*.

DOI: https://doi.org/10.7554/eLife.44649.026

tracer labeling and morphological characteristics, as well as the firing properties. Our data showed that the connection probability from non-EGIns to PCs was significantly higher in EGIn-EYFP mice (control) than in EGIn-ablated mice (DTR) (two-tailed Fisher's exact test, p=0.008; *Figure 7D*), indicating that EGIns ablation reduces inhibitory synaptic formation from non-EGIns to PCs at the early postnatal stage. In contrast, the strength and half-width of non-EGIns→PCs uIPSCs did not exhibit significant change (amplitude, 26.2 ± 9.6 pA for control, n = 13, 45.5 ± 11.7 pA for DTR, n = 8, two-tailed unpaired *t*-test, p=0.223; half-width, 18.5 ± 2.2 ms for control, n = 13, 19.1 ± 2.7 ms for DTR, n = 8, two-tailed unpaired *t*-test, p=0.872) (*Figure 7E and F*). We further assessed the presynaptic release probability from non-EGIns to PCs by analysis of failure rate, and the coefficient of variation (C.V.), and paired-pulse ratio (PPR) (*Guan et al., 2017*). Although there were no significant differences in failure rate (control, 16.9 ± 5.5%, n = 13; DTR, 10.0 ± 5.0%, n = 8; Mann Whitney *U* test p=0.399; *Figure 7G*) and C.V. (control, 0.30 ± 0.05, n = 12; DTR, 0.38 ± 0.05, n = 8; Mann Whitney *U* test, p=0.133; *Figure 7H*), PPR was significantly smaller in EGIn-ablated mice than in EGIn-EYFP mice (two-way ANOVA, $F_{(1, 72)}$=8.01, p=0.006; *Figure 7I and 7J*), suggesting that EGIns ablation reduces probability of presynaptic transmitter release from non-EGIns to PCs.

Together, these results suggest EGIns regulate synaptic formation and presynaptic transmitter release from non-EGIns to PCs at the early postnatal stage.

## Discussion

The pioneer interneurons that are generated earliest are a unique subpopulation of cortical interneurons. Although it has long been postulated that they are essential for the proper development of neural circuits, direct evidence unraveling the role of these cells in regulating circuit development in the early postnatal neocortex was elusive. Our findings suggest that pioneer interneurons in the neocortex, by contributing to neuronal synchrony at early postnatal stages, could play an important role in the wiring of immature cortical circuits.

Using transgenic mouse lines to label neurons based on their embryonic temporal origin, we observed ~60% of EGIns were SST-expressing interneurons, and the proportion was almost twice that of pRIns. Unlike perisomatic-targeting PV interneurons, SST-expressing interneurons preferentially target distal dendrites of pyramidal neurons, and precisely control the efficacy and plasticity of glutamatergic inputs (*Higley, 2014*; *Yavorska and Wehr, 2016*). Accumulating evidence suggests that SST-expressing interneurons play important roles in cortical circuit development (*Fee et al., 2017*; *Liguz-Lecznar et al., 2016*; *Tuncdemir et al., 2016*). Consistent with these studies, the hippocampal network contains a major functional subset of early-born SST-expressing interneurons with long-range projections that orchestrate immature network synchronization (*Picardo et al., 2011*; *Villette et al., 2016*). However, at present, whether such neurons exist in the developing neocortex remains unknown.

Furthermore, we found that EGIns exhibit mature morphological and electrophysiological properties during the early postnatal stages, such as complex and widespread axonal and dendritic morphologies, short AP width, and low input resistance. Moreover, EGIns received a high frequency of mEPSCs and mIPSCs and had high local synaptic connectivity with pyramidal neurons in the immature neocortex. It is worth noting that although EGIns share common properties with previously identified hub interneurons in the hippocampus (*Picardo et al., 2011*; *Villette et al., 2016*), they exhibit distinct differences. First, EGIns transiently acquired their remarkable morphophysiological attributes at early postnatal stages, and exhibited comparable properties to pRIns with respect to their morphology and electrophysiology at adolescence (P15–20). Second, EGIns received a significant increase in both excitatory and inhibitory synaptic inputs compared with pRIns at early postnatal stages. Taken together, these distinct properties of EGIns might contribute to their higher propensity to support the emergence of network oscillations and regulate cortical circuit development.

Indeed, we found that a subpopulation of EGIns can single-handedly influence network dynamics. Two distinct network oscillations, cortical early network oscillations (cENOs) and cortical giant depolarizing potentials (cGDPs), were observed in the developing neocortex (*Allène et al., 2008*). We noted two features of the network oscillations which were reflected in recording layer five neurons during P5–10: (i) the frequency is ~0.1 Hz, (ii) the synchronicity duration is 200–300 ms, which suggest that the network oscillations we recorded in neocortical layer five during P5–10 are mainly cGDPs (*Allène et al., 2008*; *Ito, 2016*; *Rheims et al., 2008*). We observed ~1/6 of EGIns in

neocortical layer five could independently alter the frequency of spontaneous network synchronization when stimulated. The proportion is much lower than reported in the hippocampus and entorhinal cortex (*Mòdol et al., 2017*; *Picardo et al., 2011*). It is worth further studying the proportion of such EGIns in other cortical layers (e.g., neocortical layer six and subplate). Moreover, unlike hippocampal hub interneurons (*Picardo et al., 2011*), we did not observe a single EGIn in neocortical layer five that could obviously trigger GDPs when stimulated. These differences might be due to distinct brain structures, or methodological differences between this (patch-clamp recording) and previous studies (multineuron calcium imaging) (*Li et al., 1994*; *Mòdol et al., 2017*; *Namiki et al., 2013*; *Picardo et al., 2011*; *Radnikow et al., 2015*). Interestingly, we found that the alteration in GDP frequency induced by stimulating EGIns tended to switch from increase at P5–7 to decrease at P8–10. The exact reason is unknown. One possibility is that GABAergic responses undergo a switch from being excitatory to inhibitory during postnatal development (*Ben-Ari et al., 2007*), but further experimentation will be needed to establish this. Furthermore, it is important to consider that EGIns are a heterogeneous population. It is also unclear whether the subpopulation of EGIns that can single-handedly alter network dynamics represents one or more morpho-physiological subtypes of interneurons. However, determining the subtypes of this subpopulation of EGIns is experimentally challenging since most interneurons during the period of GDP generation have not yet developed the characteristics that identify and classify them in adulthood.

A major finding in this study is that early-generated interneurons shape synaptic wiring during the first postnatal week. We found that ablation of EGIns after P2 impaired the development of GABAergic synaptic inputs to layer five pyramidal neurons at P5–7, but did not alter their morphological and intrinsic electrophysiological properties. Although GABAergic neurons were deleted, the reduction of inhibitory synaptic inputs onto pyramidal neurons is unlikely due to the decrease in the number of GABAergic neurons for two reasons. First, EGIns are a very sparse cell population in the neocortex (*Figure 1B*) (*Picardo et al., 2011*; *Villette et al., 2016*). Second, in the absence of a similar number of pRINs, layer five pyramidal neurons received normal inhibitory and excitatory synaptic inputs. Further work will be required to determine whether the absence (or silence) of EGIns can influence synapse formation in other cortical layers. It nevertheless remains to be determined whether the impact of EGIns on synaptic development could persist into adulthood.

Although our study clearly demonstrates the importance of EGIns in regulating inhibitory synapse formation, the precise mechanisms underlying this regulation remain largely unknown. We speculate that one potential mechanism could involve EGIns in early postnatal stages that facilitate synchronized activity, which in turn promotes inhibitory synapse formation. Indeed, we found that the absence of EGIns, but not pRINs, could significantly reduce the frequency of GDPs in the first postnatal week. Moreover, in agreement with our speculation, previous studies suggest that alteration of GDP dynamics at early postnatal stages can modulate synaptic efficacy (*Al-Muhtasib et al., 2018*; *Griguoli and Cherubini, 2017*; *Kasyanov et al., 2004*; *Mohajerani et al., 2007*; *Vargas et al., 2013*). Nevertheless, it is important to note that using current genetic strategy, EGIns were ablated not only in the neocortex but also in other brain regions (e.g., thalamus, striatum, hippocampus, etc.). Thus, we cannot exclude the possibility that certain defects of synaptic development in the neocortex may relate to the alteration of afferent inputs from these brain regions.

In summary, our study complements and expands on previous works (*Bonifazi et al., 2009*; *Mòdol et al., 2017*; *Picardo et al., 2011*; *Tuncdemir et al., 2016*) by providing new insights into EGIns that regulate network oscillations and are critical for shaping the development of precise synaptic circuits in the neocortex during early postnatal stages. A deep understanding requires investigations into the mechanisms by which EGIns orchestrate network synchronization at both the molecular and cellular level as well as how these neurons sculpt inhibitory connectivity during development. Given that disruptions of GABAergic circuitry at several points can contribute to neurodevelopmental disorders, results from this study may be particularly important for our understanding of cell-type-specific network dysfunctions in these disorders.

## Materials and methods

**Key resources table**

*Continued on next page*

*Continued*

| Reagent type (species) or resource | Designation | Source or reference | Identifiers | Additional information |
|---|---|---|---|---|
| Reagent type (species) or resource | Designation | Source or reference | Identifiers | Additional information |
| Strain, strain background (*Mus musculus*) | Dlx1/2-creER | PMID: 21867885 | RRID: IMSR_JAX:014600 | |
| Strain, strain background (*Mus musculus*) | Rosa26-EYFP | PMID: 11299042 | RRID: IMSR_JAX:006148 | |
| Strain, strain background (*Mus musculus*) | Rosa26-iDTR | PMID: 22016545 | RRID: IMSR_JAX:007900 | |
| Antibody | anti-GFP, chicken polyclonal | AVES, USA | RRID: AB_10000240 | 1:1000 |
| Antibody | anti-Calretinin, goat polyclonal | Millipore, USA | RRID: AB_90764 | 1:1000 |
| Antibody | anti-GABA, rabbit polyclonal | Sigma, USA | RRID: AB_477652 | 1:500 |
| Antibody | anti-HB-EGF(DTR), goat polyclonal | R and H System, USA | RRID: AB_354429 | 1:100 |
| Antibody | anti-NPY, rabbit polyclonal | Immunostar, USA | RRID: AB_2307354 | 1:400 |
| Antibody | anti-nNOS, rabbit polyclonal | Millipore, USA | RRID: AB_91824 | 1:1000 |
| Antibody | anti-NOS, rabbit monoclonal | Sigma, USA | RRID: AB_260754 | 1:1000 |
| Antibody | anti-PV, rabbit polyclonal | Abcam, USA | RRID: AB_298032 | 1:500 |
| Antibody | anti-SST, goat polyclonal | Santa-Cruz, USA | RRID: AB_2302603 | 1:100 |
| Antibody | anti-VIP, rabbit polyclonal | Immunostar, USA | RRID: AB_572270 | 1:200 |
| Antibody | anti-BrdU, rat monoclonal | Abcam, USA | RRID: AB_305426 | 1:250 |
| Antibody | anti-cleaved caspase-3 (Asp175), rabbit polyclonal | Cell signaling, USA | RRID: AB_2341188 | 1:400 |
| Antibody | anti-Ki67, mouse monoclonal | BD Pharmingen, USA | RRID: AB_396287 | 1:500 |
| Antibody | anti-OLIG2, rabbit polyclonal | Millipore, USA | RRID: AB_570666 | 1:500 |
| Antibody | Alexa Fluor 555 Donkey anti-goat, donkey polyclonal | Life Technologies, USA | RRID: AB_2535853 | 1:250 |
| Antibody | Alexa Fluor 555 Donkey anti-rabbit, donkey polyclonal | Life Technologies, USA | RRID: AB_162543 | 1:250 |
| Antibody | Alexa Fluor 555 Donkey anti-mouse, donkey polyclonal | Life Technologies, USA | RRID: AB_2536180 | 1:250 |
| Antibody | Alexa Fluor 568 Goat anti-rat, goat polyclonal | Life Technologies, USA | RRID: AB_2534121 | 1:250 |

*Continued on next page*

*Continued*

| Reagent type (species) or resource | Designation | Source or reference | Identifiers | Additional information |
|---|---|---|---|---|
| Antibody | Alexa Fluor 647 Goat anti-rabbit, goat polyclonal | Life Technologies, USA | RRID: AB_2535864 | 1:250 |
| Software | GraphPad Prism | GraphPad Software, USA | RRID: SCR_002798 | |
| Software | Sigma Plot | Systat Software, USA | RRID: SCR_003210 | |
| Software | SPSS | IBM, USA | RRID: SCR_002865 | |
| Software | pCLAMP | Molecular Devices, USA | RRID:SCR_011323 | |
| Software | Neurolucida | MicroBrightField, USA | RRID:SCR_001775 | |
| Software | Adobe Photoshop | Adobe system, USA | RRID:SCR_014199 | |

## Animals

Mice were raised on a 12 hr light/dark cycle with food and water *ad libitum*. The day when the vaginal plug was detected was termed as embryonic day 0.5 (E0.5), and the parturition day was termed as postnatal day 1 (P1). We bought three transgenic mouse lines from The Jackson Laboratory to label and delete EGIns or pRIns: *Dlx1/2-creER* mouse (RRID:IMSR_JAX:014600), *Rosa26-EYFP* mouse (RRID:IMSR_JAX:006148), and *Rosa26-iDTR* mouse (RRID:IMSR_JAX:007900). Female *Dlx1/2-creER* mice were crossed with male *Rosa26-EYFP* or *Rosa26-iDTR* mice to generate offspring. We gavaged the pregnant mice at E9.5 or E13.5 post vaginal plug with tamoxifen (T-5648, Sigma, USA) dissolved in olive oil (20 mg/ml, 0.1 ml/30 g body weight) to label EGIns or pRIns. In the ablation experiment, tamoxifen concentration was lowered to 3 mg/ml when gavaging pregnant mice at E13.5 to kill equal numbers of pRIns and EGIns (*Hayashi and McMahon, 2002*). Pups were injected intraperitoneally with DT (D0564-1MG, Sigma, USA) dissolved in sterilized saline (1 ng/µl, 5 µl/g body weight) once every 8 hr to delete labeled neurons at P2–4. All animal experimental procedures were approved by the Committee on the Ethics of Animal Experiments of Fudan University Shanghai Medical College.

## Immunohistochemistry and morphological reconstruction

P5–7 or P30 mice were deeply anesthetized with 1% isoflurane mixed in 0.5–1.0 L/min oxygen before heart perfusion. Mice were transcardially perfused with cold phosphate buffered saline (PBS), followed by 4% paraformaldehyde (PFA) in PBS. Brains were removed carefully from the skull and post-fixed in PFA for 12 hr at 4°C. Brains were then rinsed with PBS five times (10 min each) and sectioned coronally at 60 µm using a Leica VT1000S vibratome (Leica, Germany). Embryonic mouse brains were harvested and fixed in 4% paraformaldehyde for 5–7 hr, followed by cryoprotection in 30% sucrose in PBS overnight. Afterwards, brains were embedded in OCT and frozen at –80°C, and sliced into 12–20 µm coronal sections using Leica CM1950 (Leica, Germany).

Slices were incubated with blocking solution (5% bovine serum albumin, 0.5% Triton X-100, and 0.05% sodium azide in PBS) for 1.5–2 hr at room temperature and then incubated with primary antibody solution (1% bovine serum albumin, 0.5% Triton X-100, and 0.05% sodium azide in PBS) for 48 hr at 4°C. Slices were then rinsed with PBST (0.1% Triton X-100 in PBS) five times (10 min each) and incubated with secondary antibody solution for 12 hr at 4°C. Slices were then rinsed with PBS five times (10 min each) and mounted before visualization. Antibody information is summarized in Key resources table. For neurobiotin histochemistry, acute brain slices were fixed in PFA overnight and rinsed with PBST five times (10 min each). After incubation in blocking solution for 1.5–2 hr, slices were incubated with antibody solution containing Cy3-Streptavidin (1:500, #016–160–084, Jackson ImmunoResearch, USA; RRID:AB_2337244) for 12 hr at 4°C. Slices were then rinsed with PBS five times (10 min each) and mounted. Images were taken using an Olympus FV1000 confocal microscope (Olympus, Japan) or Hamamatsu Nanozoomer 2.0 RS (Hamamatsu, Japan) with 0.5–1.5 µm

step size. Images were brightness, contrast, and color balanced with Adobe Photoshop (Adobe system, USA; RRID:SCR_014199). Neurons were reconstructed with Neurolucida Software (MicroBright-Field, USA; RRID:SCR_001775).

## Electrophysiological recordings

P5–20 mice were deeply anesthetized with 1% isoflurane mixed in 0.5–1.0 L/min oxygen. Brains were taken out carefully and dipped in ice-cold cutting solution containing (in mM) 120 choline chloride, 2.6 KCl, 26 $NaHCO_3$, 1.25 $NaH_2PO_4$, 15 glucose, 1.3 ascorbic acid, 0.5 $CaCl_2$, and 7 $MgCl_2$ (pH 7.3–7.4, 300–305 mOsm), bubbled with 95% $O_2$/5% $CO_2$. The brains were sectioned coronally at 300 µm using a Leica VT1000S vibratome (Leica, Germany) and incubated in artificial cerebrospinal fluid (ACSF) containing (in mM) 126 NaCl, 3 KCl, 1.25 $KH_2PO_4$, 1.3 $MgSO_4$, 3.2 $CaCl_2$, 26 $NaHCO_3$, 10 glucose (pH 7.3–7.4, 300–305 mOsm) and bubbled with 95% $O_2$/5% $CO_2$ for 1 hr. Slices were then transferred into a recording chamber containing cycled ACSF at 32–34°C, bubbled with 95% $O_2$/5% $CO_2$. Patching progress was visualized under an Olympus BX61WI (Olympus, Japan) upright microscope equipped with epifluorescence illumination, 20 × and 60 × water immersion objectives, and an evolve 512 EMCCD camera (Photometrics, USA). Glass recording electrodes (5–8 MΩ resistance) filled with an intracellular solution consisting of (in mM) 93 K-gluconate, 16 KCl, 2 $MgCl_2$, 0.2 EGTA, 10 HEPES, 2.5 MgATP, 0.5 $Na_3GTP$, 10 Na-phosphocreatine, 10 mg/ml neurobiotin (SP-1120, Vector Laboratories, USA; RRID: AB_2313575), and 0.25% Alexa Fluor 568 hydrazide (A10441, Invitrogen, USA) (adjusted to pH 7.25 and 295 mOsm) were used for whole cell patching. Recordings were acquired and analyzed using two Axon Multiclamp 700B amplifiers, Digidata 1440A (Molecular Devices, USA), and pCLAMP10 software (Molecular Devices, USA; RRID:SCR_011323). Signals were sampled at 5000 Hz with a 2000 Hz low-pass filter. Liquid junction potential and cell fast capacitance were compensated. Resting membrane potential (RMP) was monitored constantly during recording and the collected data were discarded when the fluctuation of RMP was violent. Data were also discarded when their series resistance was larger than 30 MΩ.

Cells were clamped in current clamp mode and biased to −70 mV after establishing the whole-cell configuration and then injected with accumulating depolarizing currents (each sweep lasted 1.5 s with 800 ms square depolarizing current starting at −30 pA with a step of 3 pA) to evoke action potentials. The first evoked spike was selected for AP threshold, amplitude, width and input resistance calculations. AP threshold was determined from the membrane potential at the onset of the spike. AP amplitude was measured as the difference between the threshold and the peak of the spike. AP width was measured as the duration of half amplitude. Input resistance was the slope of the linear regression of current-voltage response curve sampled from the traces with negative current injection. In the synaptic connection experiment, we performed dual-patch recordings to test if two nearby neurons have synaptic connections. The pre-synaptic neuron was patched in current clamp mode and injected with stimulating currents (20 ms, 500 pA, followed by 300 ms, 300 pA, repeated 10 times with intervals of 5 s). The post-synaptic neuron was held at −70 mV in voltage clamp mode and synaptic connection was confirmed if inward currents were detected following pre-synaptic stimulation. The excitatory neurons were identified with fluorescent tracer labeling and morphological characteristics including a large pyramidal soma and thick primary dendrites decorated with spines.

For mEPSC and mIPSC recordings, we clamped neurons at −60 mV and +10 mV, respectively, combined with a Cs-based intracellular solution containing (in mM) 121.5 cesium methanesulfonate, 7.5 CsCl, 10 HEPES, 2.5 $MgCl_2$, 4 MgATP, 0.4 $Na_3GTP$, 10 sodium phosphocreatine, 0.6 EGTA, 5 QX-314 (adjusted to pH 7.25 and 295 mOsm) as previously reported (*Allene et al., 2012*; *Oh et al., 2016*; *Yang et al., 2016*; *Yao et al., 2016*). 5 µM tetrodotoxin (TTX) was added to the ACSF bath to block sodium channels. Using this method, bath application of an AMPA receptor blocker (NBQX, 10 µM) and NMDA receptor blocker (D-APV, 50 µM) or GABA-A receptor blocker (bicuculline, 10 µM) can completely block the mEPSC (at −60 mV) or mIPSC (at +10 mV) events, respectively (*Yao et al., 2016*).

To record synchronized activity, we clamped a layer five pyramidal cell at −70 mV in voltage clamp model. Based on previous studies (*Ito, 2016*; *Wester and McBain, 2016*), only inward current with amplitude larger than 100 pA and duration between 200–300 ms was termed as a GDP signal. We performed dual patch-clamp to test whether stimulating an EYFP⁺ neuron can change the frequency of GDPs recorded in a layer five pyramidal neuron. The entire experimental procedure was

divided into three consecutive phases: the pre-stimulus phase, the stimulation phase and the post-stimulus phase. EYFP$^+$ neurons were kept in current clamp mode and biased to $-70$ mV in the pre- and post-stimulus phases while injected with stimulating currents (0.1 Hz, 200 ms, 100–200 pA) in the stimulation phase. An EYFP$^+$ neuron was considered to alter the frequency of GDPs only when the GDP intervals of the stimulation phase were statistically different from that of both the pre-stimulus phase and the post-stimulus phase. We discarded the data when the GDP intervals of the pre-stimulus phase and the post-stimulus phase were statistically different. The GDP intervals were measured using Clampfit 10.6 (Molecular Devices, USA; RRID:SCR_011323).

## Statistics

GraphPad Prism 5 (GraphPad Software, USA; RRID:SCR_002798), Sigma Plot 10.0 (Systat Software, USA; RRID:SCR_003210) and SPSS 24 (IBM, USA; RRID:SCR_002865) were used for data analysis. All data were checked for normality using D'Agostino and Pearson omnibus normality test or Shapiro-Wik normality test before comparison. Comparison analysis was performed using two-tailed unpaired *t*-test, Mann Whitney *U* test, one-way or two-way ANOVA, Kruskal-Wallis test and two-tailed Fisher's exact test. *Post-hoc* tests were conducted only if the *p*-value of one-way ANOVA or Kruskal-Wallis test was 0.05 or less. All the detailed test methods, the number of experiments and *p*-values are listed in the source data. Quantifications are presented as mean ± SEM. Significant difference was recognized when p-value<0.5.

## Acknowledgements

We thank Drs. Rosa Cossart, Song-Hai Shi for comments on the manuscript. This work was supported by grants from the Natural Science Foundation of China (31725012) and the Foundation of Shanghai Municipal Education Commission (2019-01-07-00-07-E00062) to Y-CY, the Foundation of Shanghai Municipal Commission of Health and Family Planning (20154Y0034) to S-QZ.

## Additional information

### Funding

| Funder | Grant reference number | Author |
|---|---|---|
| Natural Science Foundation of China | 31725012 | Yong-Chun Yu |
| Foundation of Shanghai Municipal Education Commission | 2019-01-07-00-07-E00062 | Yong-Chun Yu |
| Foundation of Shanghai Municipal Commission of Health and Family Planning | 20154Y0034 | Shu-Qing Zhang |

The funders had no role in study design, data collection and interpretation, or the decision to submit the work for publication.

### Author contributions

Chang-Zheng Wang, Conceptualization, Resources, Data curation, Software, Formal analysis, Supervision, Validation, Investigation, Visualization, Methodology, Writing—original draft, Project administration, Writing—review and editing; Jian Ma, Conceptualization, Methodology; Ye-Qian Xu, Shao-Na Jiang, Tian-Qi Chen, Formal analysis, Validation, Investigation, Visualization; Zu-Liang Yuan, Xiao-Yi Mao, Investigation, Visualization; Shu-Qing Zhang, Resources, Investigation, Visualization; Lin-Yun Liu, Resources, Investigation; Yinghui Fu, Conceptualization, Resources, Data curation, Formal analysis, Supervision, Investigation, Writing—original draft, Writing—review and editing; Yong-Chun Yu, Conceptualization, Resources, Data curation, Software, Formal analysis, Supervision, Funding acquisition, Validation, Investigation, Visualization, Methodology, Writing—original draft, Project administration, Writing—review and editing

Author ORCIDs

Chang-Zheng Wang (iD) https://orcid.org/0000-0003-4363-1710
Yinghui Fu (iD) https://orcid.org/0000-0003-4748-4498
Yong-Chun Yu (iD) https://orcid.org/0000-0002-7456-7451

Ethics

Animal experimentation: All animal experimental procedures approved by the Committee on the Ethics of Animal Experiments of Fudan University Shanghai Medical College (permit number: 20110307-049). All surgery was performed under isoflurane anesthesia and ethanol disinfection to minimize suffering.

Decision letter and Author response

Decision letter https://doi.org/10.7554/eLife.44649.029
Author response https://doi.org/10.7554/eLife.44649.030

## Additional files

### Supplementary files

• Transparent reporting form
DOI: https://doi.org/10.7554/eLife.44649.027

### Data availability

All data generated or analyzed during this study are included in the manuscript and supporting files. Source data files have been provided for Figures 1-3, 5-7.

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
