## [Decision Letter]

Thank you for sending your article entitled "Early-generated interneurons regulate neuronal circuit formation during early postnatal development" for peer review at *eLife*. Your article is being evaluated by two peer reviewers, and the evaluation has been overseen by a Reviewing Editor and Marianne Bronner as the Senior Editor.

Given the list of essential revisions, including new experiments, the editors and reviewers invite you to respond within the next two weeks with an action plan and timetable for the completion of the additional work. We plan to share your responses with the reviewers and then issue a binding recommendation.

Summary:

This manuscript by Wang and colleagues studies the role of early-born GABAergic neurons (EGIns) in the neocortex focusing on the somatosensory region. They extend to that region the original finding in the CA3 area of the hippocampus that a subpopulation of early-born GABA neurons expressing somatostatin is capable of coordinating neuronal activity during the early postnatal period of development. They use the same experimental strategy as in Picardo et al., 2011 and had paired electrophysiological recordings that demonstrate the high connectivity between these cells and glutamatergic cells. Most importantly, by selectively ablating EGIns, they show that both spontaneous network bursts and mIPSCs are reduced in frequency.

This is an interesting study. While early generated interneurons in the developing hippocampus have been shown to develop high connectivity and serve as hub cells in affecting network synchronization, whether and how early generated interneurons influence neocortical synapse and network development are largely unclear. This study provides important new insights. The authors also pointed out the differences between early generated interneurons in the neocortex and hub interneurons in the hippocampus.

Essential revisions:

1) The authors claim that ablating EGIns impacts circuit formation based on the results that mIPSC frequency is reduced when EGIns are ablated but not when PRIns are killed. They comment that this result "clearly demonstrates the importance of EGIns in regulating inhibitory synapse formation". The impact of EGIns on 'circuit formation' is not convincing and the reasons of reduced mIPSC frequency are far from clear. First, this might simply arise from the fact that EGIns are neurons with higher connectivity than PRIns at early postnatal stages, thus ablating them leads to a dramatic reduction of inhibition onto pyramidal cells, whereas the effect is weaker if neurons with lower connectivity are removed. Second, reduced mIPSC frequency could also reflect reduced release probability from inhibitory terminals, and not necessarily a decrease in the number of synaptic terminals. To claim an effect on 'circuit formation' the authors should provide structural or electrophysiological evidence that the cortical circuit is indeed rearranged upon EGIns ablation (e.g. see the following papers providing convincing evidence of rearrangement: Tuncdemir et al., 2016; Marques-Smith et al., 2016, doi: 10.1016/j.neuron.2016.01.015). In particular, experiments should rule out that the reduced inhibition onto pyramidal cells is not simply due to decreased inputs from EGIns. Demonstrating an unequivocal effect on circuit formation is not essential for publication, but at the moment this alleged effect is highly emphasized in the manuscript.

2) The authors claim that neocortical EGIns in layer 5 share many features with hippocampal EGIns, but unlike hippocampal EGIns they are not comprised of long-range projecting cells. How can the authors be sure of this from biocytin-filling of neurons in acute slices of 300 mm thickness? In these conditions, a long axon could easily be severed and go undetected.

3) Did the authors test that diphtheria toxin administration in wild-type mice does not lead to cell death? In other words, is the effect truly specific for Cre-expressing cells? (EGIns/PRIns)

4) Temporal resolution of the approach: the authors use BrdU injections to probe the temporal resolution of their approach. This is an excellent control, given the leaky expression of reporter protein, however, they should have rather injected BrdU at E13.5 on mice gavaged at E9.5 to see how much the two studied cohorts were separated.

5) The authors used the Dlx1/2-CreER line to label interneurons by administering tamoxifen at E9.5 and E13.5. While this strategy allows temporal specific labeling, it would be important to confirm the birth date of labeled interneurons, especially at E9.5. Related to this, the precise cell types (radial glial progenitors, intermediate progenitor cells, or neurons) labeled by Dlx1/2-CreER at E9.5 and E13.5 are not clear. It would be helpful to carefully characterize the original labeled cells at the embryonic stage.

6) Dlx1/2-CreER marks both MGE and CGE, which predominantly generate superficial layer interneurons. Given that the authors observed early generated interneurons in both superficial (L2/3) and deep (L5/6) layers, are there any differences in biochemical marker expression, morphology, biophysical properties, or synaptic connectivity between early generated interneurons in superficial vs. deep layers?

7) In Figure 5, is there any correlation between the effectiveness of early generated interneurons and synaptic connectivity between interneurons and PCs? Related to this, does the switch from increase to decrease in GDP frequency between P5-7 and P8-10 reflect the synaptic action of GABA switching from excitation to inhibition?

---

## [Author Response]

[Editors' note: the authors’ plan for revisions was approved and the authors made a formal revised submission.]

Essential revisions:1) The authors claim that ablating EGIns impacts circuit formation based on the results that mIPSC frequency is reduced when EGIns are ablated but not when PRIns are killed. They comment that this result "clearly demonstrates the importance of EGIns in regulating inhibitory synapse formation". The impact of EGIns on 'circuit formation' is not convincing and the reasons of reduced mIPSC frequency are far from clear. First, this might simply arise from the fact that EGIns are neurons with higher connectivity than PRIns at early postnatal stages, thus ablating them leads to a dramatic reduction of inhibition onto pyramidal cells, whereas the effect is weaker if neurons with lower connectivity are removed. Second, reduced mIPSC frequency could also reflect reduced release probability from inhibitory terminals, and not necessarily a decrease in the number of synaptic terminals. To claim an effect on 'circuit formation' the authors should provide structural or electrophysiological evidence that the cortical circuit is indeed rearranged upon EGIns ablation (e.g. see the following papers providing convincing evidence of rearrangement: Tuncdemir et al., 2016; Marques-Smith et al., 2016, doi: 10.1016/j.neuron.2016.01.015). In particular, experiments should rule out that the reduced inhibition onto pyramidal cells is not simply due to decreased inputs from EGIns. Demonstrating an unequivocal effect on circuit formation is not essential for publication, but at the moment this alleged effect is highly emphasized in the manuscript.

We thank the reviewer for raising this issue. We agree with the reviewer that based on current data, it cannot fully support the conclusion that EGIns modulate inhibitory synaptic formation from interneurons to pyramidal cells. In the revised manuscript, to rule out that the reduced inhibition onto pyramidal cells is due to decreased inputs from EGIns, we performed dual whole-cell patch-clamp recordings to simultaneously record a layer 5 interneuron (non-EGIns) and a nearby pyramidal cell at P5–7, and compared the inhibitory synaptic connectivity and transmission between EGIn-ablated mice (EGIn-DTR^+^, DT-injected mice) and EGIn-EYFP^+^ labeled mice (EGIn-EYFP^+^, DT-injected mice). We found that the connection probability from non-EGIns to PCs was significantly higher in EGIn-EYFP^+^ mice than in EGIn-ablated mice (Figure 7D). Moreover, the paired-pulse ratio of uIPSCs from non-EGIns to PCs was significantly smaller in EGIn-ablated mice than in EGIn-EYFP^+^ mice (Figure 7I and 7J). These results suggest that EGIns regulate synaptic formation and presynaptic transmitter release from non-EGIns to PCs in the early postnatal stage.

2) The authors claim that neocortical EGIns in layer 5 share many features with hippocampal EGIns, but unlike hippocampal EGIns they are not comprised of long-range projecting cells. How can the authors be sure of this from biocytin-filling of neurons in acute slices of 300 mm thickness? In these conditions, a long axon could easily be severed and go undetected.

We thank the reviewer for pointing this out. We removed this description in the revised manuscript.

3) Did the authors test that diphtheria toxin administration in wild-type mice does not lead to cell death? In other words, is the effect truly specific for Cre-expressing cells? (EGIns/PRIns)

To explore whether diphtheria toxin administration leads to non-Cre-expressing cell death, we labeled dying cells at P5 with caspase-3 and compare their density between DT-injected CD1 mice and saline-injected CD1 mice. We found that the density of caspase-3-positive cells was similar between DT-injected CD1 mice and saline-injected CD1 mice (Figure 6—figure supplement 1), indicating that the effect of DT is specific for Cre-expressing cells.

4) Temporal resolution of the approach: the authors use BrdU injections to probe the temporal resolution of their approach. This is an excellent control, given the leaky expression of reporter protein, however, they should have rather injected BrdU at E13.5 on mice gavaged at E9.5 to see how much the two studied cohorts were separated.

We thank the reviewer for this valuable point. To test the temporal resolution of the approach, we injected BrdU at E13.5 on mice gavaged at E9.5(Figure 1—figure supplement 1C). We found that few EYFP^+^ cells co-expressed BrdU at P5(Figure 1—figure supplement 1D). This result suggests that EGIns and pRIns are two temporally separated cohorts.

5) The authors used the Dlx1/2-CreER line to label interneurons by administering tamoxifen at E9.5 and E13.5. While this strategy allows temporal specific labeling, it would be important to confirm the birth date of labeled interneurons, especially at E9.5. Related to this, the precise cell types (radial glial progenitors, intermediate progenitor cells, or neurons) labeled by Dlx1/2-CreER at E9.5 and E13.5 are not clear. It would be helpful to carefully characterize the original labeled cells at the embryonic stage.

We thank the reviewer for pointing this out. As mentioned above, to test the temporal resolution of the approach, we injected BrdU at E13.5 on mice gavaged at E9.5 to see how much the two studied cohorts (EGIns and pRIns) were separated. Our results suggest that EGIns and pRIns are two temporally separated cohorts (Figure 1—figure supplement 1C and 1D).

In addition, to explore the precise cell types labeled by Dlx1/2-CreER line, we labeled EYFP^+^ positive cells with antibodies against OLIG2 and Ki67. In these experiments, OLIG2^+^/Ki67^+^, OLIG2^-^/Ki67^+^, and OLIG2^-^/Ki67^-^ cells corresponded to RGPs, IPs, and post-mitotic interneurons (INs), respectively. We found that the majority of EYFP^+^ cells at E11 and E15 were OLIG2^-^/Ki67^-^ (Figure 1—figure supplement 2), indicating that Dlx1/2-CreER line predominantly labels post-mitotic interneurons at the embryonic stage.

6) Dlx1/2-CreER marks both MGE and CGE, which predominantly generate superficial layer interneurons. Given that the authors observed early generated interneurons in both superficial (L2/3) and deep (L5/6) layers, are there any differences in biochemical marker expression, morphology, biophysical properties, or synaptic connectivity between early generated interneurons in superficial vs. deep layers?

In the revised manuscript, we systematically compared the biochemical marker expression (Figure 1—figure supplement 3), biophysical properties (Figure 2—figure supplement 1), morphology (Figure 2—figure supplement 2), and mEPSCs/mIPSCs (Figure 3—figure supplement 1) between sEGIns and dEGIns. In brief, while the proportion of sEGIns expressing CR was significantly higher than that of dEGIns, the expressions of other biochemical markers were similar between sEGIns and dEGIns. Moreover, dEGIns showed more mature electrophysiological and morphological properties than sEGIns at P5–7. Furthermore, we observed the frequency of mEPSCs and mIPSCs and the peak amplitude of mIPSCs were comparable between dEGIns and sEGIns, while the peak amplitude of mEPSCs of dEGIns was higher than that of sEGIns.

7) In Figure 5, is there any correlation between the effectiveness of early generated interneurons and synaptic connectivity between interneurons and PCs? Related to this, does the switch from increase to decrease in GDP frequency between P5-7 and P8-10 reflect the synaptic action of GABA switching from excitation to inhibition?

We did not observe the correlation between the effectiveness of early generated interneurons and synaptic connectivity between interneurons and PCs due to the limited sample size. Moreover, we did observe the switch from increase to decrease in GDP frequency between P5–7 and P8–10 temporally coincides the synaptic action of GABA switching from excitation to inhibition. However, we currently do not know whether there indeed exists the causal link between them. Further experimentations will be needed to establish this.